# Whole Genome Sequencing Revealed Inherited Rare Oligogenic Variants Contributing to Schizophrenia and Major Depressive Disorder in Two Families

**DOI:** 10.3390/ijms241411777

**Published:** 2023-07-21

**Authors:** I-Hang Chung, Yu-Shu Huang, Ting-Hsuan Fang, Chia-Hsiang Chen

**Affiliations:** 1Department of Psychiatry, Chang Gung Memorial Hospital—Linkou, Taoyuan 333, Taiwan; atlashotel117@gmail.com (I.-H.C.);; 2Department of Psychiatry, College of Medicine, Chang Gung University, Taoyuan 333, Taiwan

**Keywords:** schizophrenia, major depression, heritability, genetics, rare mutations, oligogenic

## Abstract

Schizophrenia and affective disorder are two major complex mental disorders with high heritability. Evidence shows that rare variants with significant clinical impacts contribute to the genetic liability of these two disorders. Also, rare variants associated with schizophrenia and affective disorders are highly personalized; each patient may carry different variants. We used whole genome sequencing analysis to study the genetic basis of two families with schizophrenia and major depressive disorder. We did not detect de novo, autosomal dominant, or recessive pathogenic or likely pathogenic variants associated with psychiatric disorders in these two families. Nevertheless, we identified multiple rare inherited variants with unknown significance in the probands. In family 1, with singleton schizophrenia, we detected four rare variants in genes implicated in schizophrenia, including p.Arg1627Trp of *LAMA2*, p.Pro1338Ser of *CSMD1*, p.Arg691Gly of *TLR4*, and Arg182X of *AGTR2*. The p.Arg691Gly of *TLR4* was inherited from the father, while the other three were inherited from the mother. In family 2, with two affected sisters diagnosed with major depressive disorder, we detected three rare variants shared by the two sisters in three genes implicated in affective disorders, including p.Ala4551Gly of *FAT1*, p.Val231Leu of *HOMER3*, and p.Ile185Met of *GPM6B*. These three rare variants were assumed to be inherited from their parents. Prompted by these findings, we suggest that these rare inherited variants may interact with each other and lead to psychiatric conditions in these two families. Our observations support the conclusion that inherited rare variants may contribute to the heritability of psychiatric disorders.

## 1. Introduction

Schizophrenia and affective disorders are two major psychiatric diseases without the fully elucidated pathogenesis essential for a more comprehensive treatment strategy. Both schizophrenia and affective disorders are complex disorders involving interactions between genetic and environmental factors. Genetic factors play a significant role in the pathogenesis of these two disorders. Affective disorders consist of major depressive disorder and bipolar disorders. The heritability of schizophrenia is 81% [1], while the heritability of major depressive disorder is 31–42% [2]. For bipolar disorders, the family-based heritability of bipolar disorder is 44%, while the twin-based heritability of bipolar disorder is between 60% and 90% [3]. Elucidating the genetic basis of schizophrenia and affective disorders would provide insight into the pathogenesis of schizophrenia and affective disorders.

Genome-wide association studies have identified many common single nucleotide polymorphisms (SNPs) with modest effects associated with schizophrenia and affective disorders [4,5,6,7]. However, common SNPs explain only a part of the heritability of schizophrenia and affective disorders. There is still missing heritability information for these two disorders. Recently, increasing evidence has indicated that rare genetic and genomic variants contribute to the missing heritability of schizophrenia and affective disorders. Rare variants associated with schizophrenia and affective disorders usually have significant clinical impacts and are closer to the biological pathogenesis of schizophrenia and affective disorders [6,8,9].

Rare variants associated with schizophrenia and affective disorders include chromosomal abnormalities, translocations, copy number variations (CNVs), small insertions and deletions (indels), and single nucleotide variants (SNVs) [8,10,11,12,13,14,15]. They may occur from de novo mutation in sporadic cases or be transmitted in the familial form of schizophrenia and affective disorders [12,16]. Collecting families for genetic analysis is essential to clarify the inheritance pattern of rare variants associated with psychiatric conditions [17]. Further, rare genetic variants associated with psychiatric disorders are usually personalized, specific to the affected individuals and families [18,19,20], and have pleiotropic clinical effects [15]. Identifying specific pathogenic variants related to schizophrenia and affective disorders can help establish the molecular diagnosis [21], understand the pathogenesis [22], and provide helpful genetic counseling for the affected patients and families [23].

Whole genome sequencing (WGS) is a new sequencing technology that can identify genetic variations at the genome-wide level [24]. We used this technology with family analysis to explore the genetic basis for families that joined our precision psychiatry study series. We identified several rare specific variants associated with schizophrenia [25,26], affective disorders [27,28], and intellectual disability [27,29] in our previous publications, supporting the utility of this approach. This paper reports the clinical and genetic findings in two families with schizophrenia and major depressive disorder.

## 2. Results

We obtained 30× of read depth, on average, from WGS experiments for each subject. After filtering the variants with the criteria described in the Section 4, we obtained multiple variants in several genes implicated in psychiatric conditions through a literature review. We also interpreted the clinical significance of these variants following the “Standards and guidelines for the interpretation of sequence variants: a joint consensus recommendation of the American College of Medical Genetics and Genomics and the Association for Molecular Pathology” [30].

### 2.1. Clinical Reports of Family 1

The genealogy of this family is shown in Figure 1. The female patient was in her forties. She was born at full term without unusual complications. She grew up normally, with no remarkable life events before her mental illness. She suffered from psychotic symptoms such as auditory hallucinations and delusions of persecution when she was twenty-five. She did not have disturbing behaviors or an unstable mood. Physical and neurological examinations and laboratory tests did not discover any abnormalities. She was diagnosed with paranoid schizophrenia at that time. Her parents and younger sister did not have any mental illness. After the treatment of various currently available antipsychotics, the severity of her auditory hallucinations and delusions improved, but they did not disappear completely. She feels annoyed by the persistent auditory hallucinations and residual delusions and cannot concentrate her attention. The residual hallucinations and delusions restricted her social and occupational activities. She tried repetitive transcranial magnetic stimulation therapy, but to no effect. She currently receives antipsychotics regularly at a psychiatric outpatient clinic.

### 2.2. Genetic Findings of Family 1

We obtained genomic DNA from the patient and her parents but could not obtain DNA from her younger sister for the study. We did not detect de novo pathogenic or likely pathogenic mutations associated with psychiatric disorders in the patient. Also, we did not find recessive homozygous or compound heterozygous variants associated with psychiatric disorders in this patient. Nevertheless, we detected four inherited rare variants of unknown significance in four respective genes implicated in psychiatric disorders in the patient, including p.Arg1627Trp of *LAMA2*, p.Arg691Gly of *TLR4*, p.Pro1338Ser of *CSMD1*, and Arg182X of *AGTR2*. These variants were classified as having unknown significance according to the ACMG standards and guidelines [30]. We verified the authenticity of these variants using Sanger sequencing and show the results in Figure 2. The p.Arg691Gly of *TLR4* was inherited from her unaffected father, while the other three variants were inherited from her unaffected mother. The detailed genetic information of these four rare variants is listed in Table 1, including genomic positions, allele frequencies, and functional prediction.

### 2.3. Clinical Reports of Family 2

The genealogy of family 2 is shown in Figure 3. Both sisters were in their sixties. The elder sister was married and had a simple life until 54 years old. She suffered from a depressed mood with suicidal ideation and anxiety. She was diagnosed with major depressive disorder. Her younger sister was also married and had a simple life until she was fifty. She was diagnosed with a major depressive disorder, like her elder sister. Both sisters responded well to anti-depressants and anxiolytic treatment. Both sisters had unaffected sons who were in their thirties, respectively. They did not join this study. Their father was deceased, and their mother was in her eighties. Unfortunately, we could not obtain her agreement to join this study. Hence, we could not obtain her DNA for the family study.

### 2.4. Genetic Findings of Family 2

After analyzing the WGS data for these two sisters, we did not find de novo pathogenic or likely pathogenic mutations associated with psychiatric conditions in these two sisters. Also, we did not identify homozygous or compound heterozygous mutations related to psychiatric disorders in these two sisters. Nevertheless, we detected three rare variants shared by two sisters in three respective genes implicated in affective disorders, including p.Ala4551Gly of *FAT1*, p.Val231Leu of *HOMER3*, and p.Ile185Met of *GPM6B*. These variants were classified as having unknown significance according to the ACMG standards and guidelines [30]. We verified the authenticity of these variants using Sanger sequencing and show the results in Figure 4. These three variants were assumed to be inherited from their parents. The detailed genetic information of these three rare variants is summarized in Table 2, including genomic positions, allele frequencies, and functional predictions.

## 3. Discussion

These two families received CNV screening via chromosomal microarray analysis in our previous studies [31,32], but we did not detect rare pathogenic or likely pathogenic CNVs associated with their psychiatric conditions. Hence, they were further subjected to WGS analysis. In family 1, we did not detect pathogenic or likely pathogenic mutations associated with the mental illnesses in the proband. Nevertheless, we saw four rare inherited variants in four respective genes implicated in psychiatric disorders in the proband. The p.Arg691Gly of *TLR4* was inherited from the father, while the other three variants were inherited from the mother. Both parents were unaffected. All four variants were predicted to be deleterious through in silico analysis.

The *LAMA2* gene encodes the laminin subunit alpha 2, a member of the laminin protein complex. Laminin is an extracellular protein and a basement membrane member that interacts with other extracellular matrix proteins and mediates cellular attachment, migration, and organization during embryonic development. Pathogenic mutations of the *LAMA2* gene usually cause autosomal recessive congenital muscular dystrophy [33]. Also, mutations of *LAMA2* have been found in patients with severe childhood epilepsy [34] and malformations of cortical development [35]. Recently, a novel frameshift homozygous variant (p.Tyr1313LeufsTer4) in the *LAMA2* gene was found in a patient with congenital muscular dystrophy who had autism-like behavior [36]. A genetic study of schizophrenia discovered that recurrent de novo mutations of *LAMA2* were found significantly more often in schizophrenia than controls [37], suggesting that *LAMA2* variants may be involved in neuropsychiatric disorders. The complete human LAMA2 protein structure is not available yet in the protein data bank of UniProt or in the AlphaFold protein structure database. Hence, it is hard to predict the impact of the Arg1627Trp mutant on the LAMA2 protein structure. We also searched for interacting factors with the LAMA2 in BioGRID and found a total of 16 proteins/genes interacting with LAMA2. Among them, SNAPIN physically interacts with LAMA2 and is implicated in the pathophysiology of schizophrenia [38,39,40]. Hence, we considered that the Arg1627Trp of *LAMA2* detected in this family might contribute to the mental condition of the patient.

*CSMD1* encodes the CUB and Sushi multiple domains 1 protein, a transmembrane protein that controls several cellular functions. *CSMD1* has been linked to several diseases, such as cancer, Parkinson’s disease, and schizophrenia [41], indicating that *CSMD1* dysfunction has various clinical effects. Several genetic studies reported that *CSMD1* was associated with schizophrenia [42,43,44,45]. However, this finding was not replicated in a Chinese sample [46]. *CSMD1* is also one of the genes shared by several major psychiatric disorders such as schizophrenia, bipolar disorder, major depressive disorder, autism spectrum disorder, attention deficit hyperactivity disorder, anxiety disorder, and posttraumatic stress disorder [47,48,49,50]. Further, mRNA and protein levels of the *CSMD1* were significantly lower in the peripheral blood in schizophrenia, compared with controls [51,52,53]. These data support the idea that *CSMD1* is likely a risk gene for major psychiatric disorders, including schizophrenia. According to the UniProt, the Pro1338Ser mutation is located at the CUB domain 8 of the CSMD1. Also, six proteins/genes interact with CSMD1, according to BioGRID. Among them, the BCL6 protein level in the cerebral spinal fluid was associated with an increased risk of schizophrenia in a study [54], supporting the idea that Pro1338Ser mutation might contribute to the mental illness of the patient in this family.

The *TLR4* gene encodes the Toll-like receptor 4 protein, a member of the Toll-like receptor family. The Toll-like receptor family recognizes pathogen-associated molecules expressed on the infectious pattern and mediates the immune response. The dysregulation of the Toll-like receptor-mediated inflammation is considered a pathogenetic mechanism of schizophrenia [55,56,57]. Increased TLR4-mediated immune responses have been observed in patients with schizophrenia [58,59]. *TLR4* expression in peripheral blood cells is significantly higher in drug-naïve schizophrenia patients than in controls [60]. Further, genetic polymorphisms of *TLR4* have been associated with schizophrenia in several studies [59,61,62]. According to the UniProt, the Arg691Gly mutation is located at the TIR domain of TLR4. The TIR domain mediates interaction with the *NOX4* [63]. A study showed significant increased expression of *NOX4* in patients with schizophrenia [64]. These data support the idea that the p.Arg691Gly of *TLR4* found in this study may contribute indirectly to the mental illness of the proband in family 1.

*AGTR1* and *AGTR2* are essential in developing kidney and urinary tract systems. *AGTR1* and *AGTR2* encode angiotensin II receptor type 1 and type 2, respectively. Both receptors are key components of the rennin-angiotensin system (RAS), which regulates blood pressure and electrolytes [65,66]. The aberrant function of RAS is essentially associated with hypertension and cardiovascular diseases [67,68]. Loss-of-function mutations of *AGTR1* and *AGTR2* contribute to the congenital anomalies of the kidney and urinary tract (CAKUT) [69]. Mutations of *AGTR1* result in renal tubular dysgenesis, an autosomal recessive form of congenital kidney maldevelopment [70,71]. Nevertheless, increasing evidence indicates that aberrant RAS has expanding roles in health and diseases, including in psychiatric disorders [72,73,74].

Notably, mutations of *AGTR2* were associated with X-linked intellectual disability (XLID) [75]. In that study, the authors detected the absence of the expression of *AGTR2* in a female patient with an intellectual disability who had a balanced translocation between X and seven chromosomes. They further screened *AGTR2* mutations in 590 male patients with intellectual disability and identified one frameshift and three missense mutations of *AGTR2* in 8 patients. Thus, they suggested that *AGTR2* might play a role in brain development and cognition [75]. Other studies further supported the association between *AGTR2* mutations and intellectual disability. A study examined a sample of Finnish male patients with non-specific intellectual disabilities. The study discovered two missense mutations of *AGTR2* in their subjects [76]. Also, a novel missense mutation of *AGTR2* was identified in a Japanese male patient with severe intellectual disability, pervasive developmental disorder, and epilepsy [77]. Nevertheless, several other studies argued against the role of *AGTR2* in intellectual disability [78,79,80,81]. The discrepancies between different studies need further clarification, but these findings indicate that *AGTR2* may play a role in brain function. The proband in this family did not have hypertension, urinary tract dysfunction, or any intellectual disability except for schizophrenia. Thus, we suggest that schizophrenia might be one of the clinical phenotypes of *AGTR2* mutations. The integral AGTR2 protein is a seven-transmembrane protein. According to the UniProt, the Arg182X mutation leads to a truncated protein of AGTR2, missing two extracellular domains, three transmembrane domains, and two cytoplasmic domains. Hence, the Arg182X mutant might affect its binding to angiotensin and other interacting factors. We further searched its interacting factors in BioGRID and found nine proteins/genes interacting with the AGTR2. Among them, *PARP1* and *ZBTB16* were implicated in the pathogenesis of schizophrenia. Parp1-deficient mice showed defective neurogenesis and maldevelopment of the brain, and manifested schizophrenia-like behavior [82,83]. Zbtb16-deficient mice showed social impairment, repetitive behaviors, risk-taking behaviors, and cognitive impairment [84]. A further mechanistic study found that the *Zbtb16* transcriptome included genes involved in neocortical maturation and autism spectrum disorder, and schizophrenia pathobiology [84]. These data support the involvement of the Arg182X mutation of the ARTR2 in the patient’s mental condition.

In family 2, we also did not detect pathogenic or likely pathogenic mutations in the two affected sisters. Instead, we saw three rare variants that might be relevant to major depressive disorder, including p.Ala4551Gly of *FAT1*, p.Val231Leu of *HOMER3*, and p.Ile185Met of *GPM6B*. These three variants were present in the two sisters. Hence, these variants were assumed to be inherited from one of their parents. These variants may be present in the germline cells of their father or mother, or they may exist in mosaicism in the germline cell of their parents. However, we were not able to confirm these possibilities.

*FAT1* gene encodes the FAT atypical cadherin 1 protein and belongs to the cadherin superfamily. FAT1 has an important function in the maintenance of organs and development. Also, it activates several signaling pathways. The aberrant dysfunction of *FAT1* is primarily associated with tumor formation and growth [85,86]. However, these two sisters and their parents had no tumor history. Two studies reported that *FAT1* is a susceptibility gene for bipolar disorder [87,88], although this association was not replicated in the other study [89]. Other studies reported that *FAT1* was implicated in autism spectrum disorders [90,91,92]. These data suggest that genetic variants of *FAT1* might be associated with psychiatric conditions. According to UniProt, FAT1 is a single-transmembrane protein consisting of 4588 amino acids. The Ala4551Gly mutation is located near the cytoplasmic C-terminal of the FAT1. Interestingly, UniProt also showed that FAT1 has binary interaction with the HOMER3 protein, in which we also found a rare mutation in these two sisters.

The *HOMER3* gene encodes the homer scaffold protein 3, a member of the homer family of postsynaptic scaffold proteins. Homer family proteins mediate postsynaptic signaling and consist of *HOMER1, -2, and -3* [93]. *HOMER1* variants are associated with neuropsychiatric disorders, such as schizophrenia, Alzheimer’s disease, addiction, pain, intellectual disability, and traumatic brain injury [94]. *HOMER*2 is associated with alcohol and stress [95,96]. Also, *HOMER2* is essential in normal human hearing; *HOMER2* mutations are associated with hearing loss in humans [97,98]. There are few works in the literature about the relationship between *HOMER3* and neuropsychiatric disorders. Studies have shown that the detection of anti-HOMER3 autoantibodies is associated with autoimmune cerebellar ataxia and impaired cognition [99,100]. Nevertheless, a whole-exome sequencing study of 184 samples with major depressive disorder discovered five cases carrying rare missense variants of *HOMER3* [101], suggesting that rare genetic variants of *HOMER3* might be associated with major depressive disorder. The detection of the rare *HOMER3* mutation Val231Leu in this study is line with their findings. The HOMER3 protein consists of 361 amino acids. The Val231Leu mutation is located at the coiled coil domain of the HOMER3 protein, according to the UniProt. As mentioned before, HOMER3 and FAT1 have binary interaction; hence, both the Ala4551Gly mutation of *FAT1* and the Val231Leu mutation of *HOMER3* might contribute to the major depression of these two sisters.

The *GPM6B* is an X-linked gene encoding the glycoprotein M6B, a member of the proteolipid protein family. *GPM6B* is expressed in neurons, oligodendrocytes, and astrocytes [102,103], but the physiological function of *GPM6*B is unclear. A genome-wide association study discovered that an SNP rs6528024 of *GPM6B* was significantly associated with delay discounting [104], a trait underlying impulse control. Higher delay discounting is associated with attention-deficit/hyperactivity disorder, schizophrenia, major depression, smoking, personality, cognition, and body weight. All these data suggest the possible involvement of *GPM6B* in neurocognitive traits.

A genetic study of suicide and the X-chromosome reported an association of several markers at the X-chromosome with male suicide completers. The study also found six differentially expressed genes in the postmortem brains of suicide completers, including *GPM6B*. *GPM6B* was significantly down-regulated in the depression suicide completers compared to normal subjects [105]. Another study also discovered the reduced expression of *GPM6B* in the hippocampus of depression suicides, providing further support for the involvement of *GPM6B* in the pathogenesis of major depression [106]. UniProt states that GPM6B is a 265-amino acid protein with four transmembrane domains. The Ile185Met mutation is located between transmembranes 3 and 4. BioGRID showed 24 proteins/genes interacting with GPM6B in humans. Among them, *SLC6A4* encodes the serotonin transporter that transports the serotonin from the synaptic cleft to the presynaptic neuron. Genetic variants of *SLC6A* have been the focus of association studies of major depressive disorder and anti-depressant response [107,108]. Hence, the Ile185Met mutation of *GPM6B* might be associated with major depressive disorder through interactions with *SLC6A4*.

This study did not find pathogenic or likely pathogenic mutations associated with psychiatric disorders in these two families. However, this study detected several rare variants in genes implicated in neuropsychiatric disorders. This observation suggests an oligogenic model of psychiatric disorders. Rare variants associated with psychiatric conditions may occur from de novo mutation or be transmitted within families [12,109,110,111]. We calculated the chance of the simultaneous presence of multiple rare variants in the general population by multiplying the allele frequency of each rare variant obtained from the databases of Taiwan Biobank and Allele Frequency Aggregator (ALFA). For family 1, the estimated concurrent frequency of four rare variants (p.Arg1627Trp of *LAMA2*, p.Pro1338Ser of *CSMD1*, p.Arg691Gly of *TLR4*, and Arg182X of *AGTR2*) was zero using either the Taiwan Biobank database (0.00033 × 0 × 0 × 0.005020 = 0) or the ALFA database (0 × 0 × 0.000172 × 0.000035 = 0). For family 2, the estimated concurrent frequency of three rare variants (p.Ala4551Gly of *FAT1*, p.Val231Leu of *HOMER3*, and p.Ile185Met of *GPM6B*) in the general population was very rare using data from the Taiwan Biobank (0.017469 × 0.001980 × 0.000670 = 2.32 × 10^−7^), or zero when using ALFA (0.00029 × 0.0738 × 0 = 0). These data suggest that the opportunity for the concurrence of multiple rare variants associated with psychiatric disorders in the general population is extremely rare. Still, intermarriage may increase their coexistence.

Our observations are consistent with a recent study that reported five rare heterozygous variants associated with schizophrenia in a multiplex family. These variants were transmitted from their unaffected parents. Hence, the authors proposed that these variants had cumulative and threshold effects on the development of schizophrenia [112], suggesting that gene–gene interaction is an essential mechanism of oligogenic model psychiatric disorders [113]. Our recent paper also discovered seven rare damaging variants in a patient diagnosed with intellectual disability, autism spectrum disorder, and psychosis. All these rare variants were inherited from his unaffected parents [29].

The reporting of multiple rare inherited variants in two families affected with schizophrenia and major depressive disorders supports the idea that rare inherited variants from various genes might be a genetic mechanism that may explain a part of the missing heritability of mental disorders. These multiple genes might have a moderate risk compared to common SNPs with small and rare mutations with large effect sizes. When moderate-risk genes work together, they may reach a threshold and lead to clinical presentations. Detecting these rare oligogenic variants would provide new insights into the pathogenesis of psychiatric disorders and improve the clinical care of patients in the future.

Our study has several limitations. First, the prediction of functional impacts of variants identified in this study was inconsistent in different software. Second, we did not conduct functional assays to measure the effects of variants identified in this study at biochemical or molecular levels. Hence, we do not know their actual impacts. Third, the variants identified in the sisters of family 2 were assumed to be inherited from their parents, but we could not confirm this through family study. Fourth, the study is limited by the small sample size. Hence, our report can only be considered to show preliminary and descriptive observations. The ways in which the oligogenic model contributes to the missing heritability of psychiatric disorders needs further large-scale study.

## 4. Materials and Methods

### 4.1. Subjects

All subjects were residents of Taiwan. We consecutively recruited families with single or multiple cases affected with major psychiatric disorders into our precision psychiatry study series. The Review Board of the Chang Gung Memorial Hospital—Linkou approved this study, with the approval number 201801385A3. Each participant signed their informed consent after a full explanation of this study. We interviewed each participant and reviewed their medical records to collect their clinical information. The psychiatric diagnosis followed the diagnostic criteria of the DSM-5 (Diagnostic and Statistical Manual of Mental Disorder—5th edition) [114]. We prepared the genomic DNA from a blood sample for each participant.

### 4.2. Whole-Genome Sequencing (WGS) Analysis

We performed short-read paired-end whole-genome sequencing using the Illumina NovaSeq6000 platform (Illumina, San Diego, CA, USA) to search for small indels and SNVs. The experiment followed the standard protocols provided by the manufacturer. After a quality check, we aligned the sequencing data to the human reference genome build hg19/GRch37. Afterwards, we used the SAMtools (https://www.htslib.org/, accessed on 1 June 2022) and Genome Analysis Tool Kit (https://gatk.broadinstitute.org/hc/en-us, accessed on 1 June 2022) to refine the local alignment and generate a variant calling file (VCF) for each subject. To search for rare pathogenic SNVs associated with their psychiatric conditions in these two families, we set up several criteria to filter out the variants. We defined an allele frequency less than 0.01 as rare. Hence, we first filtered out variants with an allele frequency ≥ 0.01 from several databases. Also, we excluded variants with a read depth of less than 10×. Next, we selected variants with functional impacts of being damaging, possible damaging, probably damaging, deleterious, or disease-causing as predicted by several online software for further analysis. After obtaining a list of rare variants with compromised functional impacts, we manually curated them by reviewing the literature to determine their association with psychiatric disorders. Finally, we interpreted their clinical relevance following the “Standards and guidelines for the interpretation of sequence variants: a joint consensus recommendation of the American College of Medical Genetics and Genomics and the Association for Molecular Pathology” [30]. Further, we assessed the inheritance pattern of candidate variants by family analysis under different modes of inheritance, including de novo mutation, autosomal dominant, autosomal recessive, and X-linked inheritance. We used the SeqsLab software (https://www.atgenomix.com/seqslab-platform, accessed on 1 June 2022, Atgenomics, Taipei, Taiwan) to perform these analyses.

### 4.3. Bioinformatics Analysis

We checked the allele frequency of variants in the dbSNP (https://www.ncbi.nlm.nih.gov/snp/, accessed on 28 February 2023) and the Taiwan Biobank (https://taiwanview.twbiobank.org.tw/index, accessed on 28 February 2023). We defined variants with an allele frequency less than 0.01 as rare. We also used online software to predict the functional impacts of rare variants identified in this study, including Polyphen-2 (http://genetics.bwh.harvard.edu/pph2/index.shtml, accessed on 28 February 2023) [115], SIFT (https://sift.bii.a-star.edu.sg/, accessed on 28 February 2023) [116], and Mutation Taster (http://www.mutationtaster.org, accessed on 28 February 2023) [117]. TAfter filtering the variants with allele < 0.01, we selected variants with putatively deleterious effects and assessed their possible association with psychiatric disorders by reviewing the literature published in PubMed. To determine the damage of mutations on protein structure identified in this study, we looked at the protein structure in the UniProt Knowledgebase (https://www.uniprot.org/, accessed on 6 June 2023) [118]. Also, we examined the interacting proteins with the mutant genes using BioGrid (https://thebiogrid.org, accessed on 6 June 2023), a database of protein, genetic, and chemical interactions [119].

### 4.4. Sanger Sequencing

To verify the authenticity of mutations identified from whole-genome sequencing analysis, we designed primer pairs to obtain amplicons that covered the variants of interest using polymerase chain reaction (PCR)-based Sanger sequencing. In brief, 30 cycles of PCR were performed in a 20 μL mixture containing 100 ng DNA, 1 μM of each primer, 1X buffer, 0.25 mM of dNTP, and 0.5 U of Power Taq polymerase (Genomics, New Taipei City, Taiwan). An aliquot of the amplicon was purified and subjected to Sanger sequencing using the BigDye Terminator kit v3.1 (Applied Biosystems, Foster, CA, USA). We have summarized the primer sequences, optimal annealing temperature, and the size of each amplicon in Table 3.

## 5. Conclusions

Schizophrenia and major depressive disorders are two different diseases from clinical point of view, but several lines of study indicate shared heritability between these two disorders. Our study detected multiple inherited rare variants in two families with schizophrenia and major depressive disorder, respectively. We suggest that this might also be a common genetic mechanism shared by these two psychiatric conditions.

## Figures and Tables

**Figure 1 ijms-24-11777-f001:**
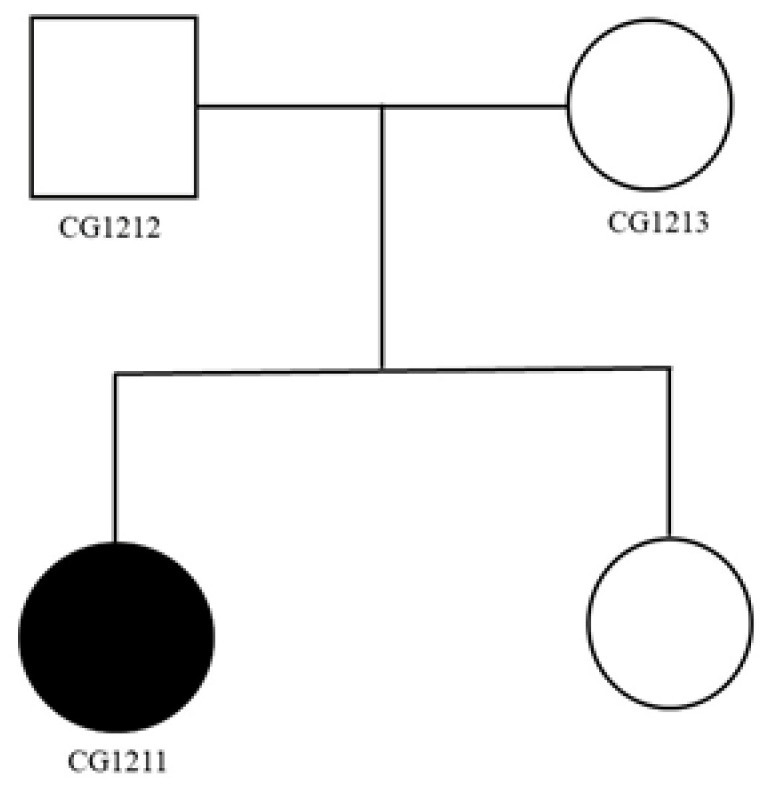
Genealogy of family 1. Black color indicates the diagnosis of schizophrenia. White color indicates the absence of mental disorder.

**Figure 2 ijms-24-11777-f002:**
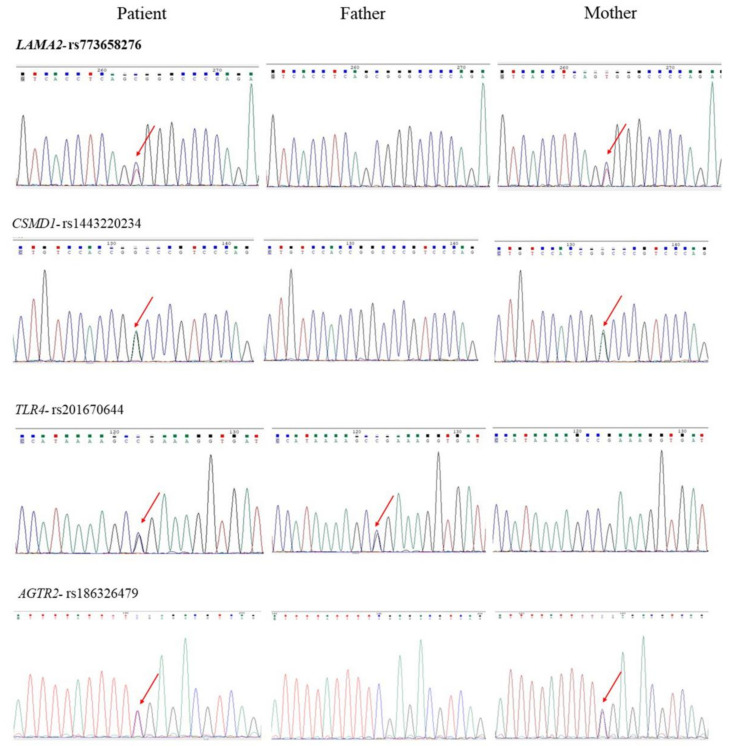
Sanger sequencing results of four inherited variants identified in family 1. Red arrows indicate the positions of nucleotide changes.

**Figure 3 ijms-24-11777-f003:**
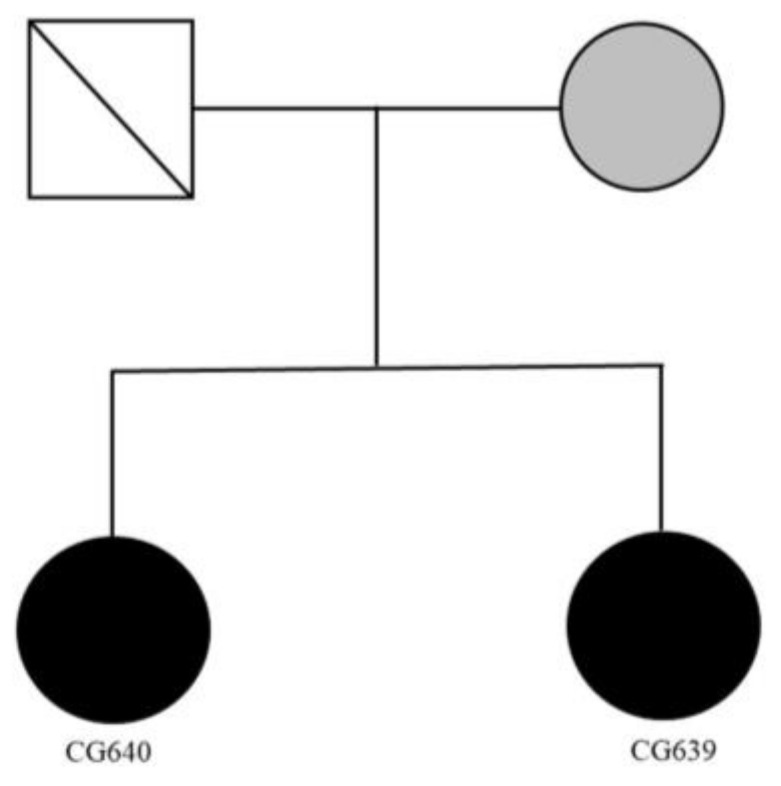
Genealogy of the family with two sisters affected by major depressive disorder and anxiety (black circle). Their mother’s mental condition was unknown (gray circle). White color indicates the absence of mental disorder.

**Figure 4 ijms-24-11777-f004:**
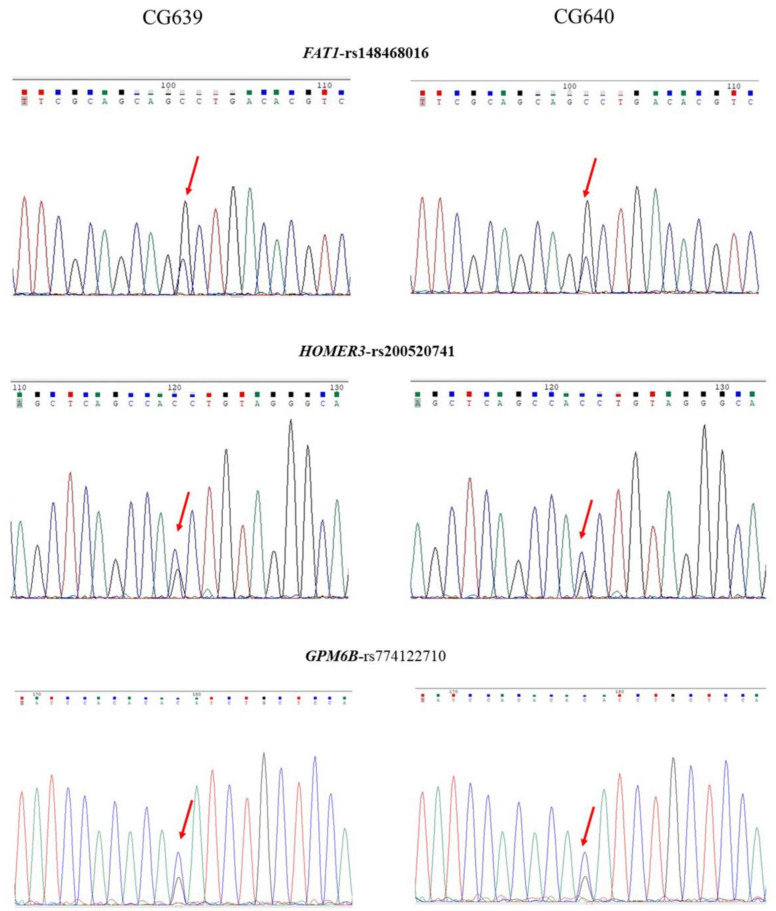
Sanger sequencing results of three inherited variants identified in family 2. Red arrows indicate the positions of nucleotide changes.

**Table 1 ijms-24-11777-t001:** Multiple inherited rare variants identified in family 1.

Gene Name and SNP	Mutation Location	Inheritance	Taiwan Biobank (MAF)	ALFA	SIFT	PolyPhen-2	Mutation Taster
*LAMA2* rs773658276	NC_000006.11:g.129691055C > T NM_000426.3:c.4879C > T NM_000426.3:p.Arg1627Trp	Mother	0.000330	0	Damaging (0.004)	Probably damaging (0.996)	Disease-causing
*CSMD1* rs1443220234	NC_000008.10:g.3141807G > A NM_033225.5:c.4012C > T NM_033225.5:p.Pro1338Ser	Mother	0	0	Damaging (0.034)	Possibly damaging (0.513)	Disease-causing
*TLR4* rs201670644	NC_000009.11:g.120476597C > G NM_003266.3:c.2071C > G NM_003266.3:p.Arg691Gly	Father	0	0.000172	Damaging (0.000)	Probably damaging (1.000)	Disease-causing
*AGTR2* rs186326479	NC_000023.10:g.115304077C > T NM_000686.4:c.544C > T NM_000686.4:p.Arg182X	Mother	0.005020	0.000035	N/A	N/A	Disease-causing

MAF: minor allele frequency; ALFA: allele frequency aggregator; SIFT: sorting intolerant from tolerant; PolyPhen-2: polymorphism phenotyping v2. N/A: not applicable.

**Table 2 ijms-24-11777-t002:** Multiple inherited rare variants identified in family 2.

Gene Name and SNP	Mutation Location	Taiwan Biobank (MAF)	ALFA	SIFT (Score)	PolyPhen-2	Mutation Taster
*FAT1* rs148468016	NC_000004.11:g.187509861G > C NM_005245.3:c.13652C > G NM_005245.3:p.Ala4551Gly	0.017469	0.00029	Damaging (0.009)	Probably damaging (1.000)	Disease-causing
*HOMER3* rs200520741	NC_000019.9:g.19042434C > G NM_001145722.1:c.691G > C NM_001145722.1:p.Val231Leu	0.001980	0.000378	Damaging (0.027)	Probably damaging (0.998)	Disease-causing
*GPM6B* rs774122710	NC_000023.10:g.13797959G > C NM_005278.3:c.555C > G NM_005278.3:p.Ile185Met	0.000670	0	Damaging (0.007)	Probably damaging (0.992)	Disease-causing

MAF: minor allele frequency; ALFA: allele frequency aggregator; SIFT: sorting intolerant from tolerant; PolyPhen-2: polymorphism phenotyping v2.

**Table 3 ijms-24-11777-t003:** Sequences of primers, optimal annealing temperature (Ta), and amplicon size for PCR-based sequencing of variants identified in this study.

Gene Name and SNP	Forward Primer Sequences (5′-3′)	Reverse Primer Sequences (5′-3′)	Size (bp)
*AGTR2* rs186326479	CTGGCTCTTTGGACCTGTGATGTG	CATTAAGGCAATCCCAGCTGACCA	316
*CSMD1* rs1443220234	CGCTGTGCCACCTACTGGAGAACT	GGGTTGTGTGAAAGCGAAATGAGC	317
*LAMA2* rs773658276	CCCCCATAGAGCTGTTGTGAAA	TACCCTGGTCAGCAGCTCGTTCAT	375
*TLR4* rs201670644	TCAAGCCAGGATGAGGACTGGGTA	CCTGAGCAGGGTCTTCTCCACCTT	300
*GPM6B* rs774122710	CCTCCCTGAAGTTTCCACCCAGAA	CTGGCTGGGTGTGTTTGGTTTCTC	320
*FAT1* rs148468016	GGGGAGTTGAGAGTCAGACTTCCG	AGAGAACCCCATGCCCCTTACCCG	244
*HOMER3* rs200520741	GCGAGGCCCAGGAACCACACTTG	TGGGTTTGAGACAATGCCAGCCTC	347

## Data Availability

The raw data are available upon request of the corresponding author.

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
