# Peer review of "Whole Genome Sequencing Revealed Inherited Rare Oligogenic Variants Contributing to Schizophrenia and Major Depressive Disorder in Two Families"

_ijms, 2023, doi:10.3390/ijms241411777_

Round 1

Reviewer 1 Report (New Reviewer)

In this work, the authors analyzed whole genome sequencing (WGS) of two families with schizophrenia and major depressive disorder, identifying 7 rare variants that are likely inherited and potentially pathogenic. The findings from this work could help the scientific community better understand schizophrenia, but more analyses are needed to better support the claims. 

Schizophrenia is known to be a disorder that arises from the malfunctioning of groups of genes, and it is thus crucial to identify which genes contribute to the pathogenesis of schizophrenia and the relationships among those genes. While the authors mentioned that gene-gene interaction could be an important mechanisms, more in-depth analyses should be performed to reveal the underlying pathways affected and to further support the claims:

1) What are genetic and physical interactors of mutated genes identified in this study? Do you identify known schizophrenia genes among the interactors? The authors could leverage existing protein-protein interaction networks (e.g., BioGRID, BioPlex, PCNet) for analysis.

2) How do the identified mutations structurally damage the protein? If these genes don't have structures in Protein Data Bank, the authors could look at AlphaFold for analysis.

The overall quality of English Language is high with minor edits needed like in line 245 ("Out" -> "Our"?).

In addition, the authors could improve the clarity in the legends. For examples, what does the values for "Taiwan Biobank" mean in Tables 1 and 2.

Author Response

performed to reveal the underlying pathways affected and to further support the claims:

  • What are genetic and physical interactors of mutated genes identified in this study? Do you identify known schizophrenia genes among the interactors? The authors could leverage existing protein-protein interaction networks (e.g., BioGRID, BioPlex, PCNet) for analysis.

Reply: Thanks for the suggestion. We follow the suggestion and check the interaction proteins/genes of every mutated gene using BioGrid or UniPro database. We identified several interacting proteins/genes associated with schizophrenia or major depression. Based on these new findings, we expanded the Discussion and marked them in red.

2) How do the identified mutations structurally damage the protein? If these genes don't have structures in Protein Data Bank, the authors could look at AlphaFold for analysis.

Reply: We check the protein structure of every mutated genes in protein database of UniPro and AlphaFold, and described their locations in the protein structure to improve the quality of our manuscript. The revised parts in the Discussion were marked in red.

 Comments on the Quality of English Language

The overall quality of English Language is high with minor edits needed like in line 245 ("Out" -> "Our"?).

Reply: We have corrected typo error, thanks.

In addition, the authors could improve the clarity in the legends. For examples, what does the values for "Taiwan Biobank" mean in Tables 1 and 2.

Reply: We have added “minor alle frequency (MAF)” to Taiwan Biobank in the legends of Table 1 and Table 2.

Reviewer 2 Report (New Reviewer)

This is a well-written study regarding schizophrenia and MDD.

Some minor remarks could be amended in order to improve quality of the work:

- lines 58-62 should be supported with adequate reference, and explained in better manner 

-  more emphasis in the Discussion should be put into clinical implications that can be derived from the results of this investigation

Author Response

- lines 58-62 should be supported with adequate reference, and explained in better manner 

Reply: We have rewritten this paragraph and added more reference as suggested.

-  more emphasis in the Discussion should be put into clinical implications that can be derived from the results of this investigation

Reply: We also add more clinical implications of our findings in the revised Discussion as suggested.

Reviewer 3 Report (New Reviewer)

Deciphering precise genetic factors and understanding heredity of widespread diagnosed mental disorders such as Schizophrenia and Major Depressive Disorder is a crucial step to manage and reduce mental health risks by strategies of personalized medicine.

In this work the authors made an attempt to discover rare oligogenic variants that may constitute inherited factors for schizophrenia in one family and major depressive disorder in another family.

The authors described four four rare variants in genes which they have claimed as implicated in schizophrenia (inherited from father to one of two daughters), and three rare variants, claimed as implicated in major depressive disorder, shared by the two sisters of another family.

There are several questions:

1. Why two quite different diseases are discussed in one paper?

2. On P.10., L.304, while discussing limitations it is written: “...variants identified in the sisters of family 2 were assumed to inherit from their parents; but we could not confirm this by family study”. In the results it is indicated that both sisters were married but there is no information whether they had children who could also be affected. Also plausible explanation of how exactly these oligogenic variants could be inherited was offered.

3. The authors have used different online software to predict the functional impacts of rare variants (Methods, P.11, L.346). I recommend providing details who which purpose precisely each software was used and provide references to papers confirming reliability of each software for the declared purposes.

4. The authors show in the study limitations that they ‘’did not conduct functional assays to measure the impacts” of these variants. But about information whether these loci are located in coding on non-coding DNA, exons/introns? This information should be available.

Checking English by native speaker is recommended.

Author Response

  1. Why two quite different diseases are discussed in one paper?

Reply: Schizophrenia and major depressive disorders are two different diseases from clinical point of view, but SNP-based studies showed overlap of susceptible SNPs between these two disorders (Lancet. 2013;381:1371–9; World J Biol Psychiatry 2014;15:200–8.; Nat Genet. 2013 Sep;45(9):984-94), indicating shared heritability between these two disorders. Our study detected multiple inherited rare variants in these two families with schizophrenia and major depressive disorder, respectively. We think this might be also a common genetic mechanism shared by these two psychiatric conditions too. Hence, we report these two families together in the present paper.   

  1. On P.10., L.304, while discussing limitations it is written: “...variants identified in the sisters of family 2 were assumed to inherit from their parents; but we could not confirm this by family study”. In the results it is indicated that both sisters were married but there is no information whether they had children who could also be affected. Also plausible explanation of how exactly these oligogenic variants could be inherited was offered.

Reply: We have rewritten this paragraph and added information as suggested.

  1. The authors have used different online software to predict the functional impacts of rare variants (Methods, P.11, L.346). I recommend providing details who which purpose precisely each software was used and provide references to papers confirming reliability of each software for the declared purposes.

Reply: We have provided references to three mutation prediction software used in this study as suggested. These three prediction software have different algorithm, it is difficult for us to explain them because we are not familiar with these algorithms. We use these software simply because they were commonly used in the literature.

  1. The authors show in the study limitations that they ‘’did not conduct functional assays to measure the impacts” of these variants. But about information whether these loci are located in coding on non-coding DNA, exons/introns? This information should be available.

Reply: The rare variants identified in this study include six missense mutations and one nonsense mutation. Hence, they are all located at coding DNA of each gene. We have added more mechanistic information of the these mutations based on the protein structure of mutated genes, and their interacting proteins/genes in the revised Discussion.

Comments on the Quality of English Language

Checking English by native speaker is recommended.

Reply: We did as suggested.

Round 2

Reviewer 1 Report (New Reviewer)

The authors have well-addressed my previous comments.

This manuscript is a resubmission of an earlier submission. The following is a list of the peer review reports and author responses from that submission.

Round 1

Reviewer 1 Report

Authors has done the massive work to detect the variants in Schizoprenia

Reviewer 3 Report

The hypothesis of an oligogenic origin of psychiatric disorders is valid and of great interest. However, I believe that the performed analyses and results do not allow to test this hypothesis.

Basically, the evidence provided to support the oligogenic origin of the disease in two families relies on the presence of heterozygous functional mutations in candidate genes for psychiatric conditions. While I agree this is needed to confirm the hypothesis of an oligogenic origin, the mere presence of mutations in candidate genes is by far not enough to confirm, or even support, this. Every human genome will have rare coding genetic variants predicted to have a functional impact, and many of them in genes related to psychiatric conditions (given the high number genes that have been related to them). The authors could at least check how frequent or infrequent is the presence of combinations of rare mutations in these genes in the general populations, or in an unbiased set of candidate genes defined from the literature. I am afraid that in the present form the manuscript does not provide this or any additional evidence (segregation, functional) to support the oligogenic hypothesis to explain the disease in these two families.

Minor:

Line 72. The authors should provide more information on how they assessed the presence of large effect variants. 

Line 94. The authors should explain which thresholds or conditions did they use to discard the presence of recessive homozygous and compound heterozygous (which otherwise are found in any human genome).

The discussion provides extensive evidence from the literature supporting the role of the detected variants in psychiatric disorders. Some considerations could be added to explore how the different genes may act in combination, or if they below to the same pathway, there evidences of co-expression, physical or functional interaction, etc.

I am not a native English speaker so I refrain from commenting on grammatical correctness. I warn however on the use of the adjective “affective” without a noun at least twice (lines 14 and 48). Also, “translocation” (line 51) should be in plural.